# Drug-Resistant *Aspergillus flavus* Is Highly Prevalent in the Environment of Vietnam: A New Challenge for the Management of Aspergillosis?

**DOI:** 10.3390/jof6040296

**Published:** 2020-11-18

**Authors:** Tra My N. Duong, Phuong Tuyen Nguyen, Thanh Van Le, Huong Lan P. Nguyen, Bich Ngoc T. Nguyen, Bich Phuong T. Nguyen, Thu Anh Nguyen, Sharon C.-A. Chen, Vanessa R. Barrs, Catriona L. Halliday, Tania C. Sorrell, Jeremy N. Day, Justin Beardsley

**Affiliations:** 1Marie Bashir Institute for Infectious Diseases and Biosecurity, The University of Sydney, Sydney 2145, Australia; mydnt@oucru.org (T.M.N.D.); thuanh.nguyen@sydney.edu.au (T.A.N.); Sharon.Chen@health.nsw.gov.au (S.C.-A.C.); vanessa.barrs@cityu.edu.hk (V.R.B.); Catriona.Halliday@health.nsw.gov.au (C.L.H.); tania.sorrell@sydney.edu.au (T.C.S.); 2Oxford University Clinical Research Unit, Ho Chi Minh City 70000, Vietnam; nguyenptuyen@gmail.com (P.T.N.); lethanhvan150895@gmail.com (T.V.L.); jday@oucru.org (J.N.D.); 3Hospital for Tropical Diseases, Ho Chi Minh 70000, Vietnam; bshuonglan@gmail.com; 4National Lung Hospital, Hanoi 10000, Vietnam; ngocn4@hotmail.com; 5Tuberculosis and Lung Diseases Department, Hanoi Medical University, Hanoi 10000, Vietnam; 6Woolcock Institute of Medical Research, Hanoi 10000, Vietnam; phuong125a@gmail.com; 7Centre for Infectious Diseases and Microbiology Laboratory Services, Institute of Clinical Pathology and Medical Research, NSW Health Pathology, Westmead Hospital, Sydney 2145, Australia; 8Department of Veterinary Clinical Sciences, Jockey Club College of Veterinary Medicine and Life Sciences, City University of Hong Kong, Kowloon Tong, Hong Kong, China; 9Westmead Institute for Medical Research, Westmead, Sydney 2145, Australia; 10Centre for Tropical Medicine and Global Health, Nuffield Department of Medicine, University of Oxford, Oxford OX3 7FZ, UK

**Keywords:** *Aspergillus flavus*, azole resistance, environmental

## Abstract

The burden of aspergillosis, especially Chronic Pulmonary Aspergillosis, is increasingly recognized, and the increasing presence of azole-resistant environmental *Aspergillus fumigatus* has been highlighted as a health risk. However, a sizable minority of aspergillosis is caused by *Aspergillus flavus*, which is assumed to be sensitive to azoles but is infrequently included in surveillance. We conducted environmental sampling at 150 locations in a rural province of southern Vietnam. *A. flavus* isolates were identified morphologically, their identity was confirmed by sequencing of the beta-tubulin gene, and then they were tested for susceptibility to azoles and amphotericin B according to EUCAST methodologies. We found that over 85% of *A. flavus* isolates were resistant to at least one azole, and half of them were resistant to itraconazole. This unexpectedly high prevalence of resistance demands further investigation to determine whether it is linked to agricultural azole use, as has been described for *A. fumigatus*. Clinical correlation is required, so that guidelines can be adjusted to take this information into account.

## 1. Introduction

*Aspergillus* are ubiquitous, globally distributed environmental saprophytes. People constantly inhale *Aspergillus* spores from the environment, although disease is relatively rare and requires either an unusually high load of spores or a weakened host. Infection, when it occurs, can result in a range of diseases; globally, the most common disease is chronic pulmonary aspergillosis (CPA), which affects more than three million people per year [1]. The mortality and morbidity of CPA can be mitigated by anti-fungal therapy. However, our ability to treat CPA and other *Aspergillus* infections is threatened by drug-resistant strains emerging under selective pressure from environmental contamination with anti-fungals used in agriculture [2]. 

*Aspergillus fumigatus* is the best studied *Aspergillus* species. In most environments, it is the species most readily isolated and causes 80–90% of human aspergillosis cases [3]. There is a relatively large body of evidence on its global environmental distribution, drug resistance, and mechanisms of resistance. Over the last decade, azole-resistance has emerged in *A. fumigatus* and has been well-described internationally, with prevalence rates of 2–14% [4,5,6,7,8,9,10,11,12,13] that increase to over 30% in selected environments with heavy azole contamination, including Vietnam [14,15]. Azole resistance is most frequently conferred by mutations in the *cyp51a* gene [16]. 

In contrast, less is known about *Aspergillus* species from section *Flavi,* the second leading human pathogenic *Aspergillus*, accounting for 15–20% of infections [17]. This section contains several species complexes including *Aspergillus flavus*, *Aspergillus oryzae*, *Aspergillus tamarii*, *Aspergillus parasiticus*, *Petromyces alliaceus*, *Aspergillus nomius*, *Aspergillus qizutongi*, *Aspergillus beijingensis*, and *Aspergillus novoparasiticus* [18]. The *A. flavus* complex contains the most important human pathogens, implicated in infections ranging from CPA to fungal keratitis. It is difficult to differentiate *A. flavus sensu stricto* from other species in the complex, so diagnostic laboratories generally report isolates as *A. flavus* species complex.

Evidence from Asia, the Middle East, and Africa identifies *A. flavus* as the predominant species in clinical isolates [17]. For example, in a study of CPA patients in Pakistan, *A. flavus* was the infecting organism in 44% of cases, compared to *A. fumigatus*, which was found in 33% of cases [19]. On environmental sampling, *A. flavus* is generally amongst the top three most frequently isolated *Aspergillus* species. Again, as might be expected from the human data, it is isolated more frequently than *A. fumigatus* in some settings [20]. *A. flavus* appears well adapted to hot humid conditions [20]. Although detailed surveillance data are lacking, this raises the possibility that *A. flavus* may play an outsized role in CPA in countries of Africa and Southeast Asia, which also have high burdens of susceptible people because of high tuberculosis (TB) incidence.

Unlike *A. fumigatus*, emergence of drug resistance in *A. flavus* has not previously been documented. Clinical breakpoints for *A. flavus* have been defined for itraconazole, with sensitivity to doses <1 mg/L indicating susceptibility, and to doses >2 mg/L indicating a resistant phenotype, using European Committee on Antimicrobial Susceptibility Testing (EUCAST) methods [21]. Epidemiological cut-off values (ECV) are defined for posaconazole (0.5 mg/L), voriconazole (2 mg/L), and amphotericin B (4 mg/L) [22]. The reported prevalence of resistance to azoles in clinical isolates has been stable, generally in the range of 0–5% [23].

Due to southern Vietnam’s hot and humid climate, we hypothesized that *A. flavus* would be readily isolated from the environment. Furthermore, based on our experience with azole-resistant *A. fumigatus* in Vietnam and considering the significant environmental contamination from azoles, as a result of their poorly regulated agri-chemical use, we hypothesized that the prevalence of azole resistance in *A. flavus* would exceed the low levels reported elsewhere. 

## 2. Materials and Methods 

### 2.1. Environmental Sampling

From January to March 2019, samples were collected at 150 locations across Ca Mau—a rural province in southern Vietnam, representative of the five key land use types: national park (*n* = 30), rice farm (*n* = 30), fruit farm (*n* = 15), shrimp farm (*n* = 45), and urban residential area (*n* = 30). At each site, we collected (1) air samples with an OxoidTM Air Sampler (100 litres/minute for 10 min), with airflow directed onto a dichloran rose–bengal chloramphenicol plate, (2) soil (at a depth of 10–15 cm), and (3) decomposing leaves (via a swab) or water (if the sampling site was in a body of water). All samples were individually sealed in zip-lock bags, transported in a cool box with ice packs to the Oxford University Clinical Research Unit in Ho Chi Minh City (HCMC) within 24 h. 

### 2.2. Isolation and Identification of Aspergillus 

Soil: 5 g of soil were suspended in 15 mL of sterile saline with 1% Tween 20 and vortexed thoroughly. The soil samples were heated at 75 °C for 30 min to optimize the yield of thermo-tolerant fungi, such as *Aspergillus*, as previously described [24,25]. The treated suspension was diluted 1:10, and 100 µL of each dilution was plated onto a maltose extract agar supplemented with 100 mg/L chloramphenicol (MEAC).

Decomposing leaves swab: the swab was soaked in 9 mL of sterile saline with 1% Tween 20, which was vortexed thoroughly, and removed before centrifuging. The resulting pellet was re-suspended in 200 µL of sterile distilled water and diluted serially 10-fold (up to 10^−2^); 100 µL of each dilution was plated onto MEAC.

Water: 5 mL of water was centrifuged at 10,000 rpm for 10 min to concentrate the fungal spores. The pellet was resuspended in 100 µL of sterile distilled water and plated on MEAC.

The plates were incubated at 37 °C for 2 to 4 days and inspected daily. From every sample plate, one colony representative of each morphotype consistent with *Aspergillus* section *Flavi* was selected for species-level identification by phenotype [26,27] and sequencing of the *β-tubulin* gene [28]. The recovery rates for *A. flavus* was calculated as the number of samples with at least one colony, divided by the total number of samples for that sample type, and reported as a percentage. 

### 2.3. Antifungal Susceptibility Testing 

All *A. flavus sensu stricto* isolates were tested for antifungal susceptibility using the EUCAST microdilution method (version E.DEF 9.3.2, April 2020) [29]. Itraconazole, posaconazole, voriconazole, and amphotericin B were chosen for clinical relevance. *A. flavus* ATCC 204304 and *Candida krusei* ATCC 6258 were included as quality control strains. Minimal inhibitory concentrations (MICs) for each strain were determined in triplicate. We reported *A. flavus* results as resistant/susceptible or non-wild type (NWT)/wild-type (WT) according to EUCAST antifungal breakpoints and ECVs (version 2.0, 2020) [22]. We reported resistance/NWT prevalence as percentages. 

## 3. Results

### 3.1. Recovery Rate

The three most commonly isolated *Aspergillus* species from our 450 samples (at 150 sites) were *Aspergillus niger* (99 isolates, recovery rate 22%), *A. flavus* (64 isolates, 14%), and *A. fumigatus* (54 isolates, 12%).

### 3.2. Prevalence of Anti-Fungal Resistance

Thirty-five *A. flavus* isolates were confirmed as *sensu stricto* and underwent susceptibility testing. MIC data for the quality-control strains were consistently within the EUCAST defined ranges. The prevalence of resistant/non-wild-type MICs are presented in Table 1. Table 2 shows the prevalence of resistant/non-wild type phenotypes by land-use type. MIC ranges and geometric means are shown in Table 3 (alongside results from selected recent international surveys of *A. flavus*). Table 4 shows the detailed isolate-level MIC data. Two isolates were resistant/non-wild-type to all antifungals tested.

## 4. Discussion

The anti-fungal MICs of our isolates are a cause of alarm. The geometric mean is above the non-wild-type ECV for voriconazole, posaconazole, and amphotericin B. For itraconazole, the only agent with defined clinical breakpoints, almost half of the isolates were frankly resistant. 

The MICs of our isolates are significantly higher than those in recent global reports. Table 3 highlights the differences, contrasting our results with those of comparable recent studies from 2017 in Brazil [30], 2018 in Iran [31], and 2018 in India [32]. Although not directly comparable, since results were obtained using Sensititre YeastOne, 2017 data from Europe are also presented [33].

We have not yet investigated the mechanisms underlying the decreased susceptibilities observed. In *A. flavus*, resistance is often conferred by efflux pumps, which become upregulated on exposure to azoles [32]. Our isolates may have had such exposure in the environment, since they were collected in a region of intensive agriculture where agri-chemicals are poorly regulated, and azole residues can be detected in cultivated soils (our unpublished data). However, in contrast to *A. fumigatus*, a link between agricultural azole use and resistance has not yet been found for *A. flavus*. Interestingly, we did not observe a lower prevalence of resistance in a national park compared to cultivated or urban land. However, the sample size for each land-use type was too small to speculate meaningfully on the impact of land use. Further investigation is required. This is the first study of *A. flavus* in Southeast Asia, and we have discovered an apparent hot spot for resistance. Our study should be replicated in other locations throughout Vietnam and neighboring countries in order to determine the extent of the risk in our region.

As anticipated, due to the hot and humid climate of southern Vietnam, *A. flavus* was more readily isolated from the environment than *A. fumigatus*, indicating that people are exposed to these spores. No clinical surveillance of infecting *Aspergillus* species has been conducted in Vietnam, so it is currently not possible to estimate the health impact of the unprecedented rates of resistance we have identified. Alongside understanding the distribution, mechanisms, and drivers of resistance, investigating its clinical impact through detailed multi-center surveillance must be a priority.

## Figures and Tables

**Table 1 jof-06-00296-t001:** Prevalence of resistant/non-wild-type phenotype amongst *Aspergillus flavus sensu stricto* isolates from the Mekong Delta Region of Vietnam against commonly used anti-fungal agents.

Resistant/Non-WT Pattern	ITC	POS	VRC	AmB
Resistant/non-WT (n/N)	17/35	27/35	6/35	9/35
Resistant/non-WT % (95% CI)	48.6% (31.4–66%)	77.1% (59.9–89.6%)	17.1% (6.6–33.7%)	25.7% (12.5–43.3%)

WT = wild-type; ITC = itraconazole; POS = posaconazole; VRC = voriconazole; AmB = amphotericin B; CI = confidence interval.

**Table 2 jof-06-00296-t002:** Prevalence of resistant/non-wild type phenotype amongst *A. flavus sensu stricto* isolates from the Mekong Delta Region of Vietnam by land-use type.

Land Use Type	Antifungal-Resistant Isolates/Total Isolates (%)	
	Azole-R	AmB-R
National park	8/8 (100)	3/8 (37.5)
Rice farm	0/3 (0)	1/3 (33.3)
Fruit farm	1/1 (100)	1/1 (100)
Aqua culture	12/13 (92.3)	2/13 (15.4)
Urban	9/10 (90)	2/10 (20)
All sites	30/35 (85.7)	9/35 (5.7)

Azole-R = resistance/non-WT MIC to any azole; AmB = resistant/non-WT; MIC = minimal inhibitory concentration to amphotericin B.

**Table 3 jof-06-00296-t003:** MIC ranges and geometric mean for *A. flavus sensu stricto* isolates from the Mekong Delta Region of Vietnam, compared to published MICs for environmental isolates from Brazil (*n* = 40), Iran (*n* = 79), India (*n* = 68), and Europe (*n* =1 9).

Country	ITC		POS		VRC		AmB	
	MIC Range (mg/L)	MIC GM (mg/L)	MIC Range (mg/L)	MIC GM (mg/L)	MIC Range (mg/L)	MIC GM (mg/L)	MIC Range (mg/L)	MIC GM (mg/L)
**Vietnam**	**1–8**	**1.52**	**0.5–2**	**0.91**	**1–4**	**2.16**	**2– >16**	**4**
Brazil	0.5–8	1.41	0.03–0.25	0.188	0.5–2	1.017	-	-
Iran	0.031–2	0.25	0.03–0.5	0.13	0.063–2	0.55	1–16	3.4
India	0.03–0.125	0.06	0.015–0.06	0.022	0.15–1	0.5	-	-
Europe *	0.03–0.25	-	0.06–0.125	-	0.125–0.25	-	-	-

GM = geometric mean. * These MICs were determined using Sensititre YeastOne (Thermo Fisher Scientific, Waltham, MA, USA). Bold = previously unpublished results from this project.

**Table 4 jof-06-00296-t004:** Anti-fungal MIC values of *A. flavus sensu stricto* isolates from the Mekong Delta Region of Vietnam (*n* = 35).

*A. flavus sensu stricto* Isolate ID	MIC (μg/mL)			
	ITC	POS	VRC	AmB
FL_1	1	1	2	4
FL_2	8	2	4	8
FL_3	2	1	2	2
FL_4	1	1	2	4
FL_5	2	1	2	4
FL_6	1	1	2	4
FL_7	1	1	2	4
FL_8	1	0.5	2	4
FL_9	2	1	2	8
FL_10	1	1	2	2
FL_11	1	1	2	4
FL_12	1	1	2	2
FL_13	1	1	1	4
FL_14	1	0.5	1	>16
FL_15	4	2	4	4
FL_16	2	1	2	2
FL_17	1	1	2	4
FL_18	1	1	2	4
FL_19	1	0.5	2	16
FL_20	2	2	2	2
FL_21	2	1	2	4
FL_22	4	1	4	8
FL_23	1	1	2	8
FL_24	2	1	2	8
FL_25	2	1	2	4
FL_26	1	0.5	2	2
FL_27	2	0.5	2	2
FL_28	2	1	4	4
FL_29	1	1	2	2
FL_30	2	1	4	4
FL_31	2	1	2	4
FL_32	1	0.5	4	8
FL_33	2	0.5	2	8
FL_34	1	0.5	2	4
FL_35	2	1	2	2

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
