# Peer review of "Drug-Resistant Aspergillus flavus Is Highly Prevalent in the Environment of Vietnam: A New Challenge for the Management of Aspergillosis?"

_jof, 2020, doi:10.3390/jof6040296_

Round 1

Reviewer 1 Report

The authors present an interesting study on the presence of antifungal-resistant A flavus strains in different Vietnamese environments uncovering a potenital source of risk for human health. This study is clearly written and well structured, but clarifications related to some specific points would be desirable.

-It is not stated whether the term "Aspergillus flavus" relates to A. flavus sensu stricto or to the species complex. This is relevant since different patterns of primary antifungal resistance are expected depending on the species involved (Gonçalves et al. Antimicrob Agents Chemother 2013 57:1944). Please specify or provide a more acurate species identification.

-It is not clear why the introduction puts so much stress specifically on CPA as A flavus complex is also related to other relevant pathologies, including ABPA and infections that could be more consistently related to the enviroments where the fungus was sought for (i.e: fungal keratytis related to lesions caused by plants), and for whom antifungal resistance has therapeutic implications. A more general approach would be ok.

-Line 66: Mutations in Cyp51A are the most frequently implicates in azole resistance but new mutations and new mechanisms are constantly described, so in its current form the statement looks misleading.

-Line 119: Selection criteria for AFST are not stated.

-Concentration is used for liquid samples, so we assume that the recovery rate of fungus is increased. For soil samples, in contrast, the estimated limit of detection is 500 ufc/gr of soil. Would it be the reason for the low number of positive  samples detected in farms, where the pressure of agricultural azoles is pressumedly higher?. Were the isolates predominantly isolated from soil, from water or from leaves? It is difficult to undestand why isolates are predominantly isolated in "aqua culture samples" but not on rice farms, is it is quite astoniching to find higher rates of resistant strains in urban samples than in agricultural settings. Could you provide a more extensive discussion on these points?

-A flavus complex is assumed to be intrisically not fully susceptible to ampho B, so some statements sound a little too alaming.

-Data from row numbers and percentages can be merged into one single column for tables 1 and 2.

Author Response

Please see the attachment, with thanks

Reviewer 2 Report

The authors present data on environmental A. flavus isolates from Vietnam, with a focus on susceptibility testing results. The study is timely and addresses an important issue, namely the emergence of antifungal drug resistance in potentially pathogenic moulds. IN the methods section, the authors present a nicely designed environmental sampling study with the potential to gain important new insights into ecology of opportunistic fungal pathogens. however, only few data are presented in selectively looking at very few isolates of one fungal species (A. flavus). In the end, data presented in the article cover only the suscpeptibility testing of 35 A. flavus strains. Even these data are currently not presented in an easily accessible way. From what the authors present, it has to be assumed that for all antifungals tested (including the relatively stable AmB) their MIC data deviate significantly from the WT MIC distributions published by EUCAST. While this could indicate a massive shift of their Population towards resistance, it could also shed doubt on the data.

Unfortunately there is more reason to assume the latter: Neither QC controls are reported nor any data provided which clearly prove that their testing methodology and reading provides data in line with EUCAST results. No mechanistic studies are performed and no attempts are made to explain or even describe potentially difficult phenotypic results. In fact, it is made impossible for the Reader to even quantify how often such results were detected by not giving the MIC valus for individual isolates. Importantly, some frequent resistance mechanisms, e.g. responsible for A. flavus azole resistance are known and it would have been easy, straightforward and important to test for the respective mutations (e.g. cyp51C mutations).

Based on all these drawback (i) lack of quality control for testing methodology, (ii) limited number of strains, (iii) lacking report of primary testing data in the manuscript and (iv) lack of independent confirmation or data on underlying mechanisms I have unfortunately serious doubts that the conclusion that “unprecedented rates of resistance” have been found or that the sampling area is “an apparent hot-spot for resistance” are solid enough.

  1. I am unsure how the „recovery rate“ has been determined. In the methods section, the authors state that they selected only one representative colony from the primary plates. Thus, the number of colony forming units is not taken into account. If this is the case, it should be clearly mentioned, e.g. by stating: “Importantly, the recovery rate is not related to abundance of both species in the environment as fungal load in the samples has not been quantified (see methods).”
  2. Based on their isolation strategy, the authors isolated 64 A. flavus isolates. It remains unclear, why only 35 were further studied. Given the overall low number of isolates, it would be important to extend the study to all of the isolates, especially as no rationale for selection of isolates is given.
  3. MICs for all isolates should be given in a table for transparency. Cumulative data suggest unusual phenotypes with differential resistance towards different azoles – it must be possible for any reader to access these data.

4. Reporting of the data needs to be amended – it is astonishing to see that 75% of A. flavus isolates are grouped as AmB susceptible/WT – likely due to the fact that an MIC of 4 (ECOFF) was used as a cut-off. However, in clinical practice, A. flavus isolates would be considered

Author Response

Please see attached, with thanks

Reviewer 3 Report

The manuscript is well-written addressing the significance of examining the antifungal resistance in A. flavus from South east Asia. The data presented would justify the importance of further research management of aspergillosis caused by A. flavus.

I have 2 major comments to raise about the methodology.

  • In Table 2, Given the low number of total isolates, the percentage of resistance is misleading. For example, 100% of the isolates from fruit farm is resistant to azole, when the total number of isolates is just 1. The authors should justify why only 35 isolates (out of 64 A. flavus isolates) were selected for antifungal susceptibility testing (line 131), leading to the low sample size.
  • How can heat treatment of the soil optimise the Aspergillus yield? Are the fungi still alive after the heat treatment? Aspergillus fumigatus is relatively more heat resistant among the Aspergillus species (PMID: 8217516), however, A. niger and A. flavus are less heat resistant. In this study, only around 1% of conidia (A. fumigatus, A. niger and A. flavus) was still live after heating at 60 degrees for 30 min (PMID: 16882610).

Other comments:

  • The authors should consider modifying the title of the manuscript to state the major finding of this study.
  • Line 78: Change “liter” to “L”
  • Line 110: Capitalize “l” in “μl”.

Author Response

Please see attached, with thanks
